# Regional Crustal Vertical Deformation Driven by Terrestrial Water Load Depending on CORS Network and Environmental Loading Data: A Case Study of Southeast Zhejiang

**DOI:** 10.3390/s21227699

**Published:** 2021-11-19

**Authors:** Wanqiu Li, Jie Dong, Wei Wang, Hanjiang Wen, Huanling Liu, Qiuying Guo, Guobiao Yao, Chuanyin Zhang

**Affiliations:** 1School of Surveying and Geo-Informatics, Shandong Jianzhu University, Jinan 250101, China; 24106@sdjzu.edu.cn (W.L.); yao7837005@sdjzu.edu.cn (G.Y.); 2Chinese Academy of Surveying and Mapping, Beijing 100830, China; dongjie@casm.ac.cn (J.D.); wangwei@casm.ac.cn (W.W.); wenhj@casm.ac.cn (H.W.); liuhl@casm.ac.cn (H.L.); zhangchy@casm.ac.cn (C.Z.)

**Keywords:** CORS network, GRACE, comprehensive calculation, crustal vertical deformation, terrestrial water load

## Abstract

Monitoring regional terrestrial water load deformation is of great significance to the dynamic maintenance and hydrodynamic study of the regional benchmark framework. In view of the lack of a spatial interpolation method based on the GNSS (Global Navigation Satellite System) elevation time series for obtaining terrestrial water load deformation information, this paper proposes to employ a CORS (Continuously Operating Reference Stations) network combined with environmental loading data, such as ECMWF (European Centre for Medium-Range Weather Forecasts) atmospheric data, the GLDAS (Global Land Data Assimilation System) hydrological model, and MSLA (Mean Sea Level Anomaly) data. Based on the load deformation theory and spherical harmonic analysis method, we took 38 CORS stations in southeast Zhejiang province as an example and comprehensively determined the vertical deformation of the crust as caused by regional terrestrial water load changes from January 2015 to December 2017, and then compared these data with the GRACE (Gravity Recovery and Climate Experiment) satellite. The results show that the vertical deformation value of the terrestrial water load in southeast Zhejiang, as monitored by the CORS network, can reach a centimeter, and the amplitude changes from −1.8 cm to 2.4 cm. The seasonal change is obvious, and the spatial distribution takes a ladder form from inland to coastal regions. The surface vertical deformation caused by groundwater load changes in the east–west–south–north–central sub-regions show obvious fluctuations from 2015 to 2017, and the trends of the five sub-regions are consistent. The amplitude of surface vertical deformation caused by groundwater load change in the west is higher than that in the east. We tested the use of GRACE for the verification of CORS network monitoring results and found a relatively consistent temporal distribution between both data sets after phase delay correction on GRACE, except for in three months—November in 2015, and January and February in 2016. The results show that the comprehensive solution based on the CORS network can effectively improve the monitoring of crustal vertical deformation during regional terrestrial water load change.

## 1. Introduction

Terrestrial water resources are an important component of the global water cycle [1] and are the material basis for human survival and development [2]. As it is affected by environmental factors, such as meteorology and human actions, TWS (terrestrial water storage) changes with time. The redistribution of water mass stimulates the time-variable evolution of the Earth’s gravitational field, while exerting an elastic load on the Earth’s lithosphere, causing the crust to move up or down [3,4].

CORS with satellite positioning is an effective method for the high-precision measurement of crustal deformation. Through the calculation and analysis of continuous observation data from CORS, we can not only study regional crustal deformation as a whole, but also the influence of global or local TWS load, atmospheric load, GWS (groundwater storage), and other changes on crustal deformation. In addition, TWS changes can be quantitatively inverted using GRACE gravity satellites [5,6]. The GRACE, launched in 2002, can provide effective time-variable gravity field observations and climate testing. Surface displacement can be observed in real time using the GNSS (Global Navigation Satellite System) [7,8]. GNSS positioning can be used as a high-precision monitoring technology for assessing crustal deformation. The nonlinear continuous time series derived from GNSS position coordinates can be adopted to derive the crustal deformation caused by TWS changes [9,10]. However, the nonlinear variation in the coordinate sequence derived from GNSS stations is a result of not only the influence of the surface mass load, but also systematic errors such as the thermal expansion and cold contraction of bedrock, the varying applicability of the correction model, and phase center model errors [10,11,12,13].

In recent years, the comprehensive analysis of crustal deformation by characteristics combining GNSS and GRACE data has become a focus of research. Previous studies have demonstrated the applicability of GNSS and GRACE data in the quantitative analysis of crustal deformation characteristics [10,14,15,16]. For example, Liu found that the GPS- and GRACE-derived results contain long-term changes and seasonal signals in the surface deformation sequence [4]. Then, Liu also found that the correlation between GRACE-inverted and GNSS-observed results derived from most monitoring stations is above 0.6 [17]. As such, they inferred that the vertical deformation of the surface in North China is mainly caused by groundwater changes. Wang found that the crustal vertical deformations observed by the GNSS and GRACE comprise obvious seasonal changes [18], and the correlations between them are strong. Some scholars applied the CORS network in analysis of the Three Gorges to monitor the spatial distribution of crustal deformation caused by the regional environmental load [19,20]. 

Southeast Zhejiang plays an important role in the development of the Yangtze River economic belt. There are abundant terrestrial water resources; however, due to the extreme climate [21] and human factors, TWS change is complex, and the earth surface fluctuates to different extents. Most of the current research is focused on the analysis of local meteorological changes [22,23,24,25], whereas there are relatively few studies on the influence of water load in this area. Furthermore, some think that the vertical deformation information of regional terrestrial water load can be obtained by spatial interpolation after removing the influence of environmental load from GNSS elevation sequences. However, this method cannot effectively separate the influence of the vertical deformation caused by the groundwater equilibrium effect and tectonic movements from the GNSS elevation time series.

In this paper, we present a comprehensive method to quantitatively calculate the vertical deformation of the crust as driven by terrestrial water load in southeast Zhejiang, combining data from multiple sources via the load deformation theory and the spherical harmonic analysis method. In addition, the calculation process is described in detail, and we compare its results with the GRACE-derived results. In this way, we can explore the capacity for monitoring crust vertical deformation resulting from terrestrial water load changes using a comprehensive calculation method based on the CORS network and environmental load data.

## 2. Materials and Methods

### 2.1. Data Used in Comprehensive Calculation

#### 2.1.1. CORS Network Data

We used data from 38 CORS stations resulting from continuous GNSS observations in Zhejiang Province from 2015 to 2017 [26,27,28]. The data processing was mainly carried out via GAMIT/GLOBK software. Using each station’s daily GNSS data, single-day region relaxation solutions for the station and satellite orbit were obtained. Then, we performed a whole network adjustment using GLOBK software and gained the time series of station coordinate changes under the ITRF2008 framework. Solid tides, tidal ocean signals, and tidal atmospheric signals were removed using the IERS 2010 protocol [29]. In addition, we selected 12 IGS sites to participate in the solution of the CORS data in southeast Zhejiang, including the AIRA, ARTU, BJFS, DAEJ, LHAZ, PIMO, POL2, SHAO, TCMS, WUHN, URUM, and YSSK stations. The location of CORS in southeast Zhejiang is shown in Figure 1. The statistics of all the corrections applied over the entire CORS network and all the IGS stations in the GAMIT calculation are shown in Table 1. All CORS stations use the Tianbao receiver, and these CORS stations do not belong to the IGS station. The latitude and longitude coordinates of the 12 IGS stations in the study area are beyond the scope of this paper, so they are not show in in the map. As regards the spatial locations of the IGS sites, their site names and corresponding latitude and longitude are detailed in Table 2. 

#### 2.1.2. Atmosphere Pressure Data

The monthly global atmospheric pressure model is derived from ECMWF’s reanalyzed ERA–interim surface pressure product data, with a spatial resolution of 0.25° × 0.25° (https://www.ecmwf.int/ accessed on 8 October 2020) [30]. The monthly data of regional atmospheric pressure are taken from 98 meteorological stations in Zhejiang (http://data.cma.cn accessed on 8 October 2020). The time period studies was from January 2015 to December 2017, which is the same as the period for the GNSS data. Combining the model’s data with the observation data, we calculated the crustal vertical displacement caused by atmospheric loading changes. These include non-tidal atmospheric pressure changes as well as the tide component [31].

#### 2.1.3. MSLA Data

The MSLA (monthly sea level anomalies) grid data are provided by AVISO (Archiving Validation and Interpretation of Satellite Oceanography), and the spatial resolution is 0.25° × 0.25° (https://www.aviso.altimetry.fr accessed on 8 October 2020). These data were combined with altimetry data from multiple satellites, such as TOPEX/Poseidon, Jason-1/2, and Envisa. Necessary geophysical corrections, including tidal corrections and inverse barometer corrections, were applied by AVISO [32]. The time period was from January 2015 to December 2017. In addition, we used MSLA data taken directly from the AVISO website to calculate the crustal vertical displacement as driven by non-tidal ocean load. This was calculated via Green’s function method, namely Equation (4) for analyzing the surface load. Under Green’s function method, we used Green functions with PREM, which were obtained from the public data released by Wang [33]. Non-tidal ocean loading was not significant at all sites. The range of surface vertical deformation caused by the non-tidal ocean load is −4 mm to 3 mm. 

#### 2.1.4. GLDAS Model

The monthly GLDAS global hydrological model was constructed by NASA (National Aeronautics and Space Administration) and the NCEP (National Centers for Environmental Prediction) [34], reflecting monthly changes in soil water, ice, and snow on the Earth’s surface (https://mirador.gsfc.nasa.gov accessed on 8 October 2020). Its spatial resolution is 0.25° × 0.25° and it covers the period from January 2015 to December 2017, incorporating four terrestrial surface modes—NOAH, VIC, CLM, and MOSAIC. Because this NOAH model of GLDAS hydrological model is the most common used by scholars, we adopted the monthly GLDAS NOAH land surface model L4 with 0.25° × 0.25° resolution, V2.1, to calculate the crustal vertical displacement caused by the soil water load change. No GRACE data were assimilated into the reanalysis. Soil water load (soil moisture) is different from groundwater storage, and that NOAH model does not account for groundwater. 

### 2.2. GRACE Data and Post-Processing

The latest GSM data from the GRACE gravity satellite are used in this paper, as derived from the Center for Space Research (University of Texas at Austin, CSR) in April 2018. The level-2 (RL06) monthly gravity field model GSM is regularized by the SH coefficient, which deducts the effects of solid tides, ocean tides, solid polar tides, non-tidal atmospheres, and oceanic influences, as well as gravity disturbances caused by other entities, such as the sun and the moon. The RL06 version of the GSM time series incorporates the same process of surface mass redistribution as the RL05 product.

We used GRACE RL06 data to invert the crustal vertical deformation, according to Formula (3). We used RL06 data, with a degree/order of 60. We did not truncate the degree/order 96 to degree/order 60, but directly employed the value of 60 [35,36]. The noise of the high-degree and high-order coefficients was processed by fan filtering with a smooth radius of 300 km [37,38]. The correlation error was corrected via P_3_M_15_ decorrelation filtering, and the width of the fixed sliding window was adjusted to 5 points, which means that the initial order of the GRACE gravity field coefficient decorrelation filtering should be 15, the sliding embedding dimension should be 5 points, and the fitting polynomial order should be 3. Here, the embedding dimension was 5 and the fitting order was 3. The coefficients of degree 1 were replaced by Sweason’s results [39]. In addition, we modified the expressions of GRACE C_20_ values by replacing the C_20_ values from the CSR/GFZ/JPL-RL06 GSM files with the corresponding values from TN11. The TN11 values of SLR-derived C_20_ are not interchangeable with the TN07 or TN05 values, due to differences in background models and the absence of background rates in the former. For any month without data, the coefficients were derived by averaging values from the two adjacent months. 

### 2.3. Method

If the load values *h_w_* of discrete points on the Earth’s surface are known, the radial displacement driven by this load can be calculated by the following formula:(1)$$\mu (\varphi ,\lambda ,t)=\int_{0}^{2\pi }d\lambda^{\prime }\int_{0}^{\pi }\rho_{w}h_{w}U(\psi )R^{2}\sin\varphi^{\prime }d\varphi^{\prime }$$
where μ(φ,λ,t) is the radial displacement, φ,λ is the colatitude and longitude of the calculated point, respectively, $\varphi^{\prime },\lambda^{\prime }$ is the colatitude and longitude of the load point, respectively, *t* is time, U(ψ) is Green’s functions of displacement, and ψ is the angular distance between the calculated point and the load point.

Considering the convergence of the load LOVE number, the Green function of radial displacement can be obtained as follows [40]: (2)$$U(\psi )=\frac{ah_{\infty }^{\prime }}{2M\sin(\psi /2)}+\frac{a}{M}\sum_{n=0}^{N}\left(h_{n}^{\prime }-h_{\infty }^{\prime }\right)P_{n}(\cos\psi )$$
where $h_{n}^{\prime }$ is radial load LOVE number, *a* is the average radius of the Earth, *M* is the mass of the Earth, and *P_n_* is the Legendre function.

Farrell states that if the spherical harmonic expansion of the surface load value is known, the radial displacement of the Earth caused by this load can be calculated via the following formula:(3)$$\begin{array}{cc} \mu (\varphi ,\lambda ,t) & =\frac{3}{\rho_{e}}\sum_{n=0}^{\infty }\frac{h_{n}^{\prime }}{2n+1}Q_{n}\left(\varphi ,\lambda ,t\right)\\ & =\frac{3}{\rho_{e}}\sum_{n=0}^{\infty }\sum_{m=0}^{n}\frac{h_{n}^{\prime }}{2n+1}\left[Q_{n,m}^{c}\left(t\right)\cos\left(m\lambda \right)+Q_{n,m}^{s}\left(t\right)\sin\left(m\lambda \right)\right]{\bar{P}}_{n,m}(\cos\varphi ) \end{array}$$
where μ(φ,λ,t) is radial displacement, *t* is time, φ,λ is the colatitude and longitude of the calculated point, respectively, $\rho_{e}$ is the Earth’s density, $h_{n}^{\prime }$ is the radial load LOVE number, $Q_{n}\left(\varphi ,\lambda ,t\right)$ is the load value derived from the harmonic coefficient, $Q_{n,m}^{c}$, $Q_{n,m}^{s}$ is the expanded Stokes spherical harmonic coefficient with n degree and m order, and ${\bar{P}}_{n,m}$. is the associated Legendre function.

Equation (1) depicts the convolution with the Green function and the field of mass over the Earth; the design matrix is the coefficient matrix of the Green function, which can be obtained from Equation (2). This paper follows the inversion model of Argus [41].
((*Ax* − *b*)/σ)^2^ + *β*^2^ (*L*(*x*))^2^→*min*(4)
where *b* is the vector of the GNSS observations of vertical displacement, σ is the vector of standard errors, *x* is the vector of groundwater water mass at each pixel, and *A* is the design matrix consisting of Green’s functions relating the groundwater water mass at a single pixel to the GNSS vertical observation, which can be obtained via Equation (2). *L* is the Laplacian operator, and *β* is a roughness factor specifying the degree to which the values of neighboring pixels may differ. 

Due to the limited number of GNSS observation stations in the region, the number of equations is less than the number of unknowns, resulting in rank deficiency in the coefficient matrix of the normal equation. Therefore, the inversion of terrestrial water storage using GNSS observation data is an ill-posed problem. We use the ridge method proposed by Hoerl for parameter estimation, which is a classical regularization method. 

In this paper, daily geodetic height data from 38 CORS stations were used. Based on the load deformation theory and spherical harmonic analysis method, we removed all non-groundwater load effects including atmosphere, soil moisture, and ocean loading from the monthly non-linear geodetic height time series of CORS, and the residual time series were obtained. The residual time series included the vertical displacement driven by the groundwater load and the non-load vertical displacement. Then, we inverted the GWS from the residual time series and used the GWS to recompute the crustal vertical displacement from the GWS load. Our aim was to separate the non-load vertical deformations within the CORS stations’ data from the time series of geodetic height change. The final calculation should give the groundwater storage (equivalent water height), and then the influence of regional terrestrial water load can be obtained. We call the process of calculating the unknowns of Equation (4) a comprehensive calculation. Through comprehensive calculation, the influence of terrestrial water load change could be accurately obtained for locations without stations, and regional crustal vertical deformation as driven by terrestrial water load in the study area could be monitored. The technical flow chart of our method is given in Figure 2, illustrating the main steps.

## 3. Results and Analysis

To improve the accuracy of the inversion solution, first, the influences of environmental load on the geodetic height observation, such as the atmospheric load, soil water load, and non-tidal ocean load, were removed. The observation equation was constructed using residual geodetic height, and the residual equivalent water height was then calculated, corresponding to the change in groundwater storage.

The changes in geodetic height after gross error detection are presented via the chocolate scatter series shown in Figure 3. The reconstructed time series for the FFT period is presented via the black curve in Figure 3. The method of reconstructing the FFT period is a kind of smoothing, which reduces the noise in the geodetic height time series derived from the CORS. The crustal vertical deformations caused by atmosphere load, soil water load, and non-tidal ocean load are shown as the red curve, cyan curve, and green curve, respectively, in Figure 3. We have depicted one inland site and one site near the coast to highlight the effects of the non-tidal ocean—the PCJM station and ZJWL station, respectively. The results of the other 36 CORS stations are shown in Appendix A, as a supplement to Figure 3.

To clearly see the amplitude of the change in the reconstructed geodetic height derived from the 38 CORS in the study area, we plotted and displayed them in Figure 4. The amplitude of variation of the reconstructed geodetic height (maximum minus minimum) is between 30 mm and 50 mm. From Figure 3, we can see that among the three forms of environmental load, the magnitude of the crustal vertical deformation caused by the atmospheric load is the greatest, reaching −7~7 mm. For all sites, even those near the coast, the crustal vertical deformation caused by atmospheric load is greater than that caused by soil water load and non-tidal ocean load (illustrated in the Appendix A).

In order to evaluate the quality of the time series data derived from the CORS stations, this paper comprehensively analyzed and evaluated the quality of a one-day time series, as well as the nonlinear motion stability of 38 CORS stations in southeast Zhejiang, given the set weights. Here, equally weighted observations were used in the experiment. The weights of all stations were set to 1, but if the data quality at specific stations was relatively poor, these weights were reduced accordingly. The reduced standard is based on the reciprocal of RMS for each CORS. The weights of each site are shown in Table 3.

Furthermore, we used the L-curve method to select the optimal regularization parameter, causing an ideal inversion result to be produced as quickly as possible. We used the ridge estimation method proposed by Hoerl for regularization. The high-resolution parameter approximation problem refers to the selection of the best roughness factor *β*. A roughness factor *β* results in a reasonable fit with the data. A smaller value of *β* would result in a greater variation in water thickness between neighboring pixels than is given by the uncertainties in the GNSS data. A larger value of *β* would result in lateral variations in water thickness that are insufficient to adequately fit the GNSS data [41]. The groundwater storage equivalent water is obtained via a comprehensive calculation. In this paper, the change grid for the spatial distribution of groundwater storage over 36 months, from January 2015 to December 2017, was obtained via inversion using CORS data. Taking April and August 2017 as an example, the change in the equivalent height of groundwater storage obtained via an analysis of CORS network data is shown in Figure 5. The z-axis represents the groundwater storage equivalent in mm. 

The crustal vertical displacement caused by groundwater loading is calculated via Equation (4), as shown in Figure 6. In addition, we added the vertical displacement from groundwater load with the influence of the soil water load, so the monthly terrestrial water load could be obtained, and the results of five sites from different regions are shown in the Figure 7. The results of the other 33 CORS stations are shown in Appendix A as a supplement to Figure 6.

Figure 6 shows that the crustal vertical deformation caused by groundwater load changes at all the 38 sites in the east–west–south–north–central sub-regions is consistent across the whole time scale. However, in the winter of each year from 2015 to 2017, the groundwater load changes decreases, which led to an significant increase in the crust in southeast Zhejiang; however, there were differences in the magnitude of this change in the five sub-regions. 

Figure 7 shows that the crustal vertical deformation caused by terrestrial water load has obvious seasonal characteristics. In the winter (February), the ground is clearly lifted; in the spring (May), the ground begins to sink slightly; in summer (August), the degree of ground subsidence intensifies. The ground sinking amplitude increased in the fall (November) in 2015 and 2017, but not 2016. In February and November 2015, February and August 2016, and February and May 2017, the spatial magnitude (absolute value) of the load impact presented a step-by-step distribution that gradually decreased from inland to coastal areas, which may be related to offshore ocean signals. 

To analyze the temporal characteristics of crustal vertical deformation caused by terrestrial water load changes, we provide the time series derived from the CORS in the eastern (YUEQ, TAIZ), northern (ZJXJ, PANA), western (SUIC, LONQ), and southern (RUIA, ZJJN) areas, achieved via interpolation, because the CORS are not necessarily located on the center of spatial grid. In this paper, the surface vertical deformation was obtained from the terrestrial water load. The values of deformation at the CORS positions are shown in Figure 8. The results of the other 30 CORS stations are shown in Appendix A as a supplement to Figure 8.

In order to clearly illustrate the spatial characteristics of the seasonal variations in vertical deformation resulting from terrestrial water load, we constructed Figure 7. The difference between Figure 7 and Figure 8 is that Figure 8 exhibits more time variation characteristics from January 2015 to December 2017. At the same time, in order to reflect the overall time variation in the study area, the deformation timings of the 38 stations in the east, west, south, and north areas are all displayed. The results in Figure 8 show the surface vertical deformation caused by groundwater load and soil water load. 

It can be seen from Figure 8 that, from 2015 to 2017, the crustal vertical deformation caused by changes in terrestrial water load in the eastern, northern, western, and southern areas show spatial and temporal heterogeneity. From February to March every year, the terrestrial water load decreases, which causes the Earth’s crust to rise, after which the ground surface drops sharply. This subsidence trend continues up to July–August, before decreasing. In January of most years, the crustal fluctuations are small. In addition, the crustal vertical deformation caused by terrestrial water load also shows obvious inter-annual variations. At all the 38 CORS sites, the greatest uplift of the crust occurred in February 2016, and varied from area to area. The ground upward displacements from most of the 38 CORS sites in the western area were close to 25 mm, the amplitude of which was bigger than that in the eastern area. The variations in the amplitude of vertical displacements at CORS stations in the southern and northern regions were roughly the same. The maximum subsidence occurred in the west and south, at about 15 mm.

## 4. Comparison between CORS and GRACE Results

This section comprises a test to validate our groundwater level inversion method. This is because the local hydrological features we analyzed could not be seen with GRACE, but only with CORS. Therefore, we test the use of GRACE for the verification of CORS network monitoring results.

Based on the above analysis of the temporal–spatial characteristics of the terrestrial water load effect, we see that the terrestrial water load impact in southeast Zhejiang has certain spatial–temporal heterogeneity. The signal intensity of the vertical deformation caused by the terrestrial water load in the inland region is greater than that in the coastal region. The amplitude difference in terrestrial water load deformation between the eastern, western, southern, and northern regions is at the millimeter level. Considering most of 38 CORS sites, the amplitude variation of them in the western region was the greatest, followed by the southern region, and the amplitude variation in the eastern region was the smallest. There are obvious fluctuations in the temporal distribution. In summary, the surface vertical deformation caused by changes in terrestrial water load in the study area from 2015 to 2017 showed spatial and temporal heterogeneity. For comparison, we used data from the latest version of GRACE RL06 released by CSR.

To determine the leakage error generated by the filtering process, we adopted a scale factor *k* = 1.33 for calibration, as derived from the PCR-GLOBWB model [42]. PCR-GLOBWB [43,44] is a conceptual, process-based water balance model used for the terrestrial part of the hydrological cycle (except Antarctica). It simulates, for each grid cell and each time step (daily), the water storage in two vertically stacked soil layers and an underlying groundwater layer, as well as the water exchange between the layers and between the top layer and the atmosphere (rainfall, evaporation, and snow storage) [45].

The GRACE RL06 data reach up to August 2016, so we used the monthly gravity field model from January 2015 to August 2016 to calculate the effects of terrestrial water load changes on crustal vertical deformation in southeast Zhejiang, with reference to Equations (2) and (3) in Section 2.3. In addition, we gave the same time reference as in the above data; that is, we used the average of all months in the three years. We compare the results from both models for the same period in southeast Zhejiang herein. 

To compare the temporal variation in both sets of results, we present the time series derived from the 10 CORS stations uniformly distributed in study area during the period of January 2015 to August 2016, as shown in Figure 9, and the results of the other 30 CORS are shown in Appendix A as a supplement to Figure 9. The yellow curve represents the vertical deformation from the terrestrial water load monitored by GRACE without phase delay correction. The cyan curve represents the vertical deformation from the terrestrial water load monitored by GRACE with phase delay correction. The red curve represents the vertical deformation from the terrestrial water load monitored by CORS.

Figure 9 shows that the GRACE monitoring results have a two-month phase delay compared with the CORS network at all the 38 sites. It expresses that CORS station can monitor the surface vertical displacement caused by the change of terrestrial water load in a timely manner; however, GRACE can perceive the water load deformation after 2 months. After correcting the phase delay, the correlation coefficient between the two monitoring results was greatly improved, increasing to about 0.7, as shown in Table 4. Overall, both the CORS network and GRACE detected that the crust derived from the changes in terrestrial water load fluctuation clearly. 

At the same time, the vertical amplitude of the terrestrial water load monitored by the CORS network was greater than that of GRACE. In November 2015 and January and February 2016, for example, the displacement derived from CORS was significantly greater than that inverted by GRACE. In addition, the 38 CORS stations in Figure 6 are located at different positions in the study area, and the time series of displacement monitored at every site are significantly different in amplitude. However, the GRACE monitoring results show the same variation characteristics at different locations in the study area, indicating that the CORS network can reflect local characteristics when used to monitor the vertical deformation of the terrestrial water load [18]. GRACE is limited by its spatial resolution, as it struggles to identify details at such scales as the study area [46,47,48].

Table 4 shows that the correlation coefficient of crustal vertical deformation related to terrestrial water load, as monitored by CORS and GRACE, which reaches 0.64~0.74. The results derived from the CORS network and GRACE are generally close for the same period. 

It is important to discuss the spatial resolutions of both regional GNSS positioning in the geometric space for a network of stations, as well as the GRACE observations in the geopotential space. GRACE measures the integrated regional effect of mass redistribution, which produces load effects, while GNSS explains the deformations that arise in short wavelengths when contaminated by local physical effects (which is also why the monitoring results of the two technologies are different). Each method has its own errors that should be considered in further studies.

## 5. Conclusions

In this paper, we present a comprehensive calculation method, based on CORS network and environmental load data, for quantitatively monitoring the crust vertical deformation caused by terrestrial water load in southeast Zhejiang. Through our method, the influence of changes in terrestrial water load can be accurately obtained for locations without stations, and the crustal vertical deformation process can be monitored. In addition, we analyzed the spatial–temporal change characteristics of vertical deformation as related to terrestrial water load, and compared them with GRACE’s results at all of the 38 sites. The results are as follows: The effect of terrestrial water load on the crust vertical deformation in southeast Zhejiang from 2015 to 2017 reaches the centimeter scale, the amplitude changes from −1.8 cm to 2.4 cm, and the seasonal variation is obvious. In addition, among the three kinds of environmental load, the effect on crustal vertical deformation of atmospheric load is the greatest, at −7~7 mm;The spatial distribution of the terrestrial water load effect in the study area from 2015 to 2017 takes the ladder form from the inland to coastal regions, and considering most of the 38 CORS sites, the amplitude change in the west is higher than that in the east. The surface vertical deformation caused by groundwater load change in the east–west–south–north–central sub-regions shows obvious fluctuations;The vertical deformation of terrestrial water load based on the comprehensive calculation of the CORS network can reflect spatial–temporal characteristics more precisely than GRACE. The signal strength of GRACE’s monitoring results has no significant effect on the spatial distribution, while the results derived from the CORS network show that a stronger signal in the west than in the east. As regards temporal distribution, the amplitude change in GRACE’s monitoring results over three years is significantly smaller than that of CORS, and there are significant differences between the two sets of results in individual months, especially in November 2015 and January and February 2016;GRACE’s monitoring results contain a two-month phase delay compared with the CORS network at all the 38 CORS sites. After correcting the phase delay, the correlation coefficient between the two results was significantly improved. This indicates that CORS station can respond to the vertical displacement of terrestrial water load in a more timely manner than GRACE. GRACE is limited by its spatial resolution, and it struggles to identify details at such spatial scales as those employed in the study area; there is little distinction between sites in the GRACE data. This is the essence of the value of the GNSS, as it helps us to see local effects that may not be captured by GRACE.

## Figures and Tables

**Figure 1 sensors-21-07699-f001:**
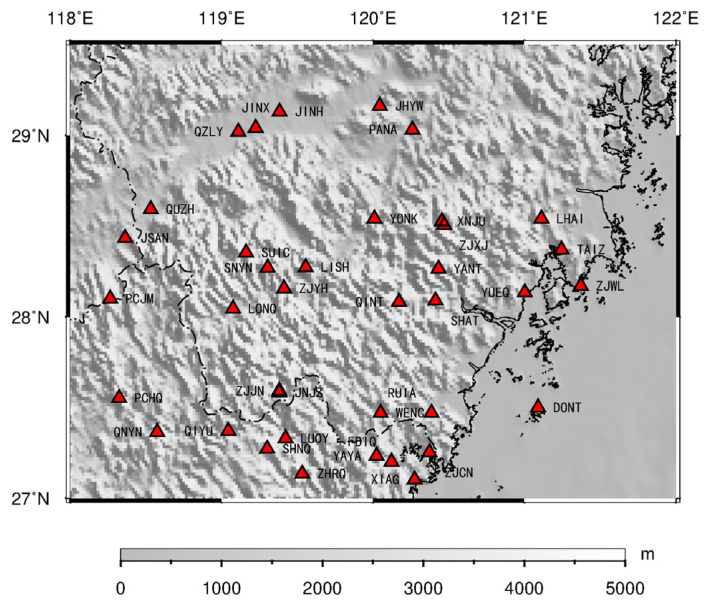
Location of CORS stations in southeast Zhejiang area.

**Figure 2 sensors-21-07699-f002:**
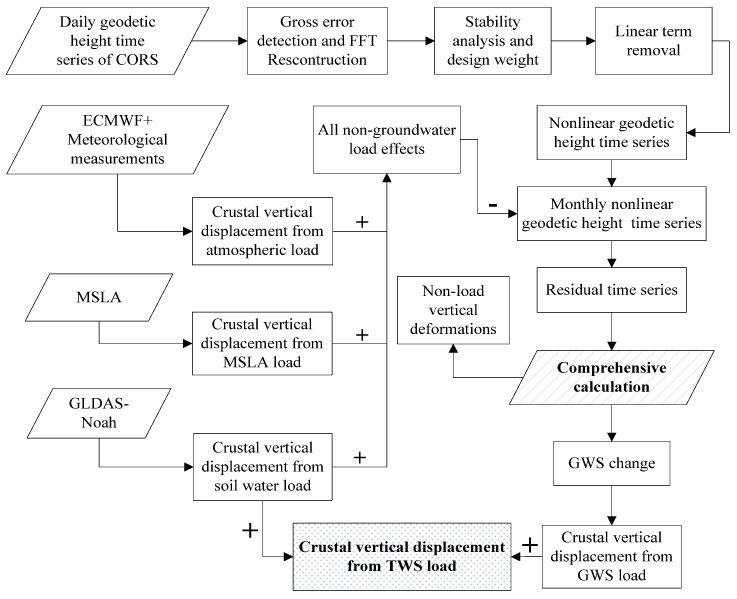
The main steps of calculating the vertical deformation of terrestrial water load based on the CORS network comprehensive calculation method.

**Figure 3 sensors-21-07699-f003:**
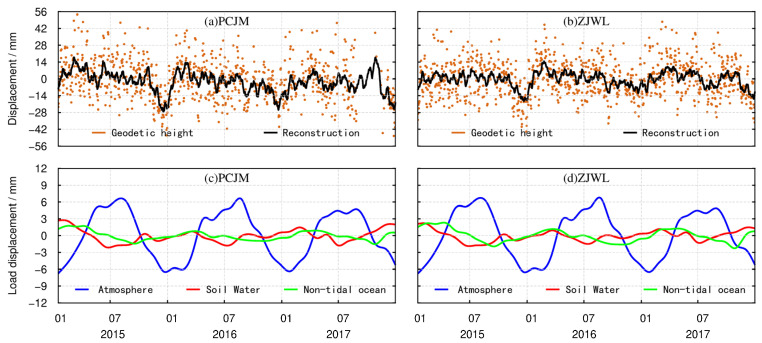
(**a**) The processed geodetic height at PCJM site; (**b**) The processed geodetic height at ZJWL site; (**c**) Crustal vertical deformation at PCJM site caused by environmental load; (**d**) Crustal vertical deformation at ZJWL site.

**Figure 4 sensors-21-07699-f004:**
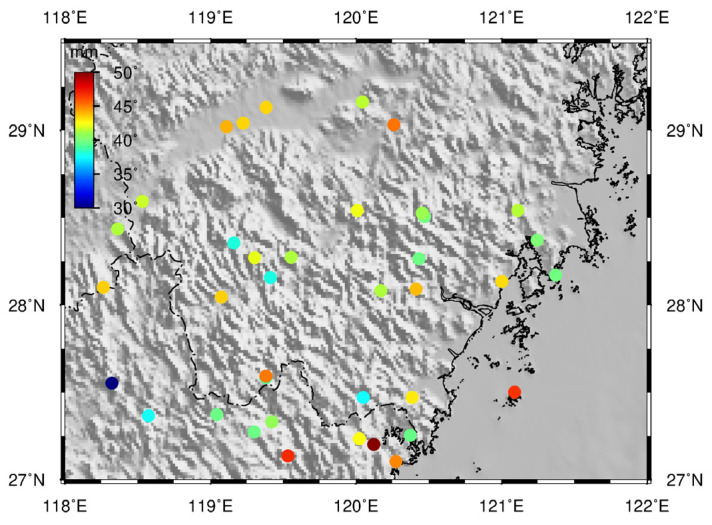
Map with colored dots depicting the amplitude of the reconstructed geodetic height for every site.

**Figure 5 sensors-21-07699-f005:**
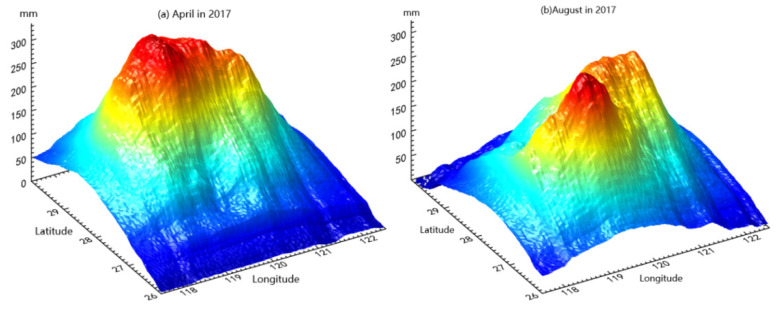
(**a**) The monthly change in groundwater storage derived from a comprehensive calculation of CORS data, taking April 2017 as the examples. (**b**) The monthly change in groundwater storage, taking August 2017 as the examples.

**Figure 6 sensors-21-07699-f006:**
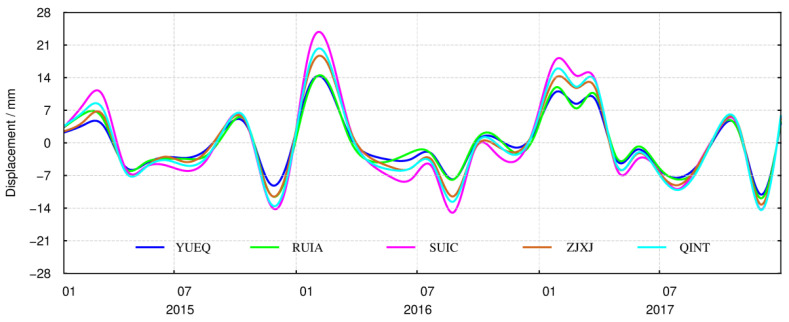
Time series of the crustal vertical deformation caused by groundwater loading at the CORS.

**Figure 7 sensors-21-07699-f007:**
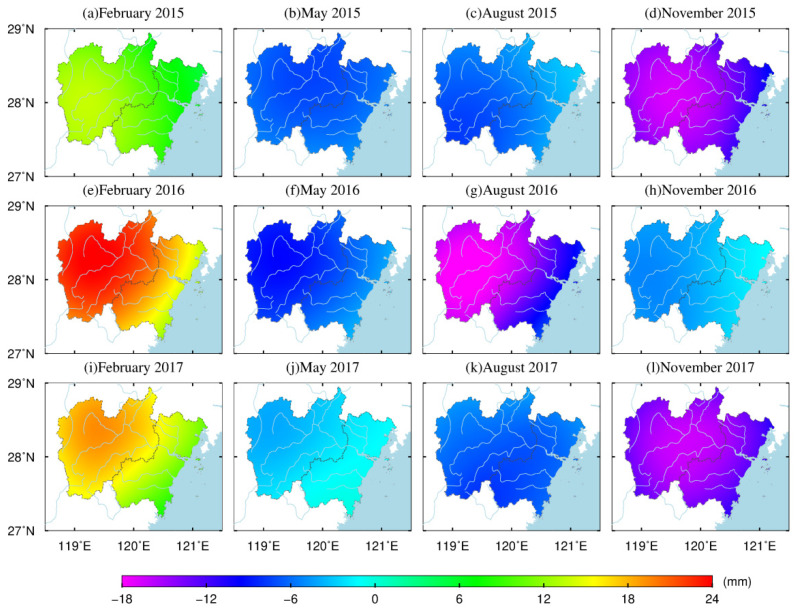
The spatial vertical deformation driven by the change in terrestrial water load from 2015 to 2017.

**Figure 8 sensors-21-07699-f008:**
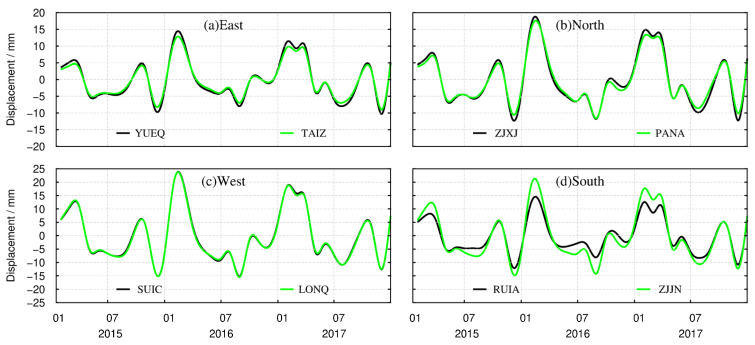
Temporal distribution of vertical deformation derived from the terrestrial water load at CORS positions.

**Figure 9 sensors-21-07699-f009:**
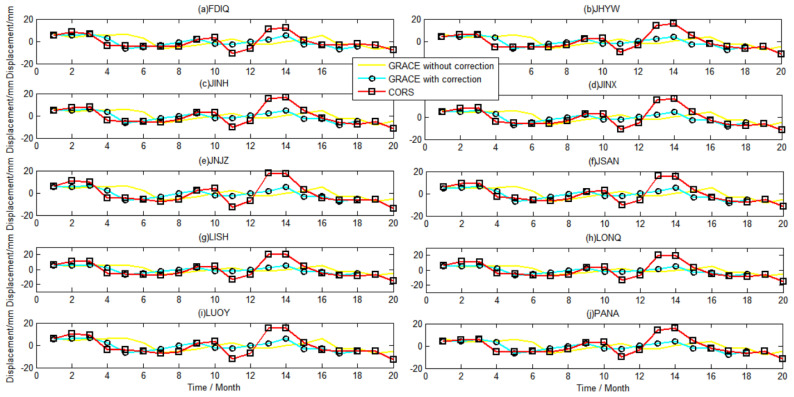
Comparison of the effects of terrestrial water load derived from the CORS network and GRACE.

**Table 1 sensors-21-07699-t001:** The main parameters used in the GAMIT calculation.

Parameter	Processing Mode
Sampling interval data	15 s
Satellite cut-off elevation angle (°)	10
Baseline processing mode	BASELINE
Ionosphere delay model	LC_AUTCLN
Satellite clock error model	Precise clock offset and orbit products of IGS
Tropospheric model	Saastamoinen + GPT2w + estimation
Solar radiation pressure model	ECOMC model
Solid tide model	IERS2010
Ocean tide model	FES2004(otl_FES2004.grid)
Atmospheric mapping function	VMF1
Inertial framework	J2000
Framework of prior coordinates	ITRF2014
PCO/PCV	IGS14 atx
Ambiguity resolution	LAMBDA method
IGS station priori coordinates	The coordinates under ITRF2008 published by SOPAC

**Table 2 sensors-21-07699-t002:** 12 IGS sites used in CORS network calculation the southeast Zhejiang area.

Site	Longitude and Latitude	Site	Longitude and Latitude	Site	Longitude and Latitude
AIRA	130.5995 31.8240	ARTU	58.5604 56.4298	BJFS	115.8924 39.6086
DAEJ	127.3744 36.3994	LHAZ	91.1040 29.6573	PIMO	121.0777 14.6357
POL2	74.6942 42.6797	SHAO	121.2004 31.0996	TCMS	120.9873 24.7979
WUHN	87.6006 43.8079	URUM	114.3572 30.5316	YSSK	142.7167 47.0297

**Table 3 sensors-21-07699-t003:** The weights 38 CORS in the study area.

Site Name	Weight	Site Name	Weight	Site Name	Weight	Site Name	Weight
DONT	0.8	LUOY	1.0	SHAT	1.0	YAYA	1.0
FDIQ	0.8	PANA	0.8	SHNQ	1.0	YONK	1.0
JHYW	1.0	PCHQ	0.8	SNYN	1.0	YUEQ	1.0
JINH	1.0	PCJM	0.8	SUIC	1.0	ZHRQ	1.0
JINX	1.0	QINT	1.0	TAIZ	1.0	ZJCN	1.0
JNJZ	1.0	QIYU	1.0	WENC	0.8	ZJJN	1.0
JSAN	1.0	QNYN	1.0	XIAG	1.0	ZJWL	1.0
LHAI	1.0	QUZH	1.0	XNJU	1.0	ZJXJ	1.0
LISH	1.0	QZLY	1.0	YANT	1.0	ZJYH	1.0
LONQ	1.0	RUIA	1.0				

**Table 4 sensors-21-07699-t004:** Correlation coefficient of temporal results of the CORS network and GRACE, with correction of phase delay.

Station Name	Value	Station Name	Value	Station Name	Value	Station Name	Value
FDIQ	0.69	JHYW	0.64	JINH	0.68	JINX	0.68
JNJZ	0.71	JSAN	0.72	LISH	0.70	LONQ	0.71
LUOY	0.71	PANA	0.64	PCHQ	0.74	PCJM	0.73
QINT	0.68	QIYU	0.72	QNYN	0.74	QUZH	0.72
QZLY	0.69	RUIA	0.68	SHAT	0.67	SHNQ	0.71
SNYN	0.71	SUIC	0.71	WENC	0.69	XIAG	0.68
YANT	0.67	YAYA	0.69	YONK	0.66	ZHRQ	0.71
ZJCN	0.68	ZJJN	0.71	ZJYH	0.71	DONT	0.67
LHAI	0.64	TAIZ	0.64	XNJU	0.66	YUEQ	0.65
ZJWL	0.64	ZJXJ	0.66				

## Data Availability

Not applicable.

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
