# Peer review of "Regional Crustal Vertical Deformation Driven by Terrestrial Water Load Depending on CORS Network and Environmental Loading Data: A Case Study of Southeast Zhejiang"

_sensors, 2021, doi:10.3390/s21227699_

Round 1
Reviewer 1 Report
My comments are attached.

Author Response
Dear Editor and reviewers:
Thanks for your valuable comments about our manuscript. We have carefully considered your reviewer comments and made changes accordingly. All revisions are highlighted in red in the manuscript. Meanwhile, a point-to-point response to comments is attached. Please see the attachment.
If any problem, please feel free to contact us.
Best regards,
--Qiuying Guo.

Reviewer 2 Report
The title is too long. I suggest that it should be shorten.
In Abstract, “ In view of the lack of spatial interpolation method based on GNSS (Global Navigation Satellite System) el-evation time series to obtain terrestrial water load deformation information, this paper proposes to use CORS (Continuously Operating Reference Stations) network combined with environmental loading data such as ECWMF atmospheric data, GLDAS (Global Land Data Assimilation System) hydrological model and mean sea level anomaly (MSLA) data, based on load deformation theory and spherical harmonic analysis method, taking 38 CORS stations in southeast Zhejiang province as an example, comprehensively solve the vertical deformation of the crust caused by regional ter-restrial water load changes from January 2015 to December 2017, and compare with GRACE (Grav-ity Recovery and Climate Experiment) satellite monitoring results.” The sentence above is too long.
The method should be described briefly in Abstract.
The abstract is too long. I suggest that it should be shorten.
As the manuscript is only 11 pages, its type should be “communication”, not article.
Author Response

(The authors gave the same response as above.)

Reviewer 3 Report
Manuscript Number: sensors-1434326
Full Title:
Monitoring of Regional Crustal Vertical Deformation Driven by Terrestrial Water Load Combining with CORS Network and Environmental Loading Data: A Case Study of Southeast Zhejiang
General comment
The paper submitted to Sensors MDPI proposed to use CORS network (38 CORS) combined with environmental loading data (ECWMF, GLDAS, MSLA), to monitoring the regional terrestrial water load deformation in People’s Republic of China (southeast Zhejiang province). The work is analysed in original submission. The paper has been edited according to the Sensors MDPI instructions for authors and is set up correctly in all its paragraphs. The study shows a considerable effort on the part of authors to implement the project and its potential applications. The content of the paper is potentially interesting, and with improvement, it might be published. However, in its current form it is not acceptable (especially for a high quality journal like Sensors MDPI) for several reasons, that follow in the specific comments.
Specific comment
With regret, the paper shows some limitations that need to be deeply resolved by the authors with a new and completely revised version of the manuscript.
In particular:
- references: usually in MDPI, bibliographic references are placed not with the author's name (Feng et al, for example) but in [ ]; likewise for internet references (http:.....). Please read the authors' instructions and correct all citations in the paper;
- dataset: the authors clearly state that they are studying a CORS network of 38 stations, but show graphs, diagrams and results of only at least 8; it is indispensable in the new version of the document that all stations are analysed, with new graphs, tables and text; please modify and clarify for the reader;
- time range: data from 2015 to 2017 are inadequate for an analysis of the load deformation, for two main reasons: 1) data are automatically collected by CORS and must therefore be extended to at least 2021; 2) scientific studies are based on at least 5-10 years of data; in my experience, data with smaller intervals can also be used, but this must be clearly specified in the text, making reference to other publications in our field, so I have inserted appropriate bibliographic references in the technical comments;
- software used: given that GAMIT/GBLOCK is one of the most used scientific programs, together with Bernese and Gipsy, it is mandatory that the authors clearly indicate for the reader all the options used (tropospheric, ionospheric models, ..., cut-off, ambiguity resolution, ...) so that your experiment can be clearly reproduceable in a scientific context;
- CORS characteristics: all CORS specifications (antenna, receiver, belonging to international networks such as IGS, or CMONC) must be indicated in a new table;
- figures: some have to be modified because they are of low quality;
- results: it is clear that in order to analyse the behaviour and performance of the other 30 stations, the results need to be extended, both with tables, diagrams, figures and above all in the text;
- conclusions: it is essential to extend the conclusions to the other 30 stations.
Technical comment
In relation to the previous points, I would like to advise the authors to deeply modify the structure of the work as follows in attachment file.
In conclusion the work deserves publication after major revisions, in my opinion. I will be available to authors for the assessment of the manuscript at the subsequent submission.
Best regards

Author Response

(The authors gave the same response as above.)

Round 2
Reviewer 1 Report
I thank the authors for making a detailed review of their paper and for their explanations.
However, I’m sorry yet that I still have some questions, see attached pdf.

Author Response

(The authors gave the same response as above.)

Reviewer 2 Report
Thanks for you careful revisions.
Author Response

(The authors gave the same response as above.)

Reviewer 3 Report
Dear Authors,
all requests for modifications of the text have been correctly fulfilled and therefore in my opion the paper is now acceptable in present form.
Best regards
Author Response

(The authors gave the same response as above.)
